# Thymoquinone-Loaded Chitosan Nanoparticles as Natural Preservative Agent in Cosmetic Products

**DOI:** 10.3390/ijms23020898

**Published:** 2022-01-14

**Authors:** María Mondéjar-López, Alberto José López-Jiménez, Joaquín C. García Martínez, Oussama Ahrazem, Lourdes Gómez-Gómez, Enrique Niza

**Affiliations:** 1Instituto Botánico, Departamento de Ciencia y Tecnología Agroforestal y Genética, Universidad de Castilla-La Mancha, Campus Universitario s/n, 02071 Albacete, Spain; maria.mondejar3@alu.uclm.es (M.M.-L.); albertojose.lopez@uclm.es (A.J.L.-J.); Oussama.ahrazem@uclm.es (O.A.); MariaLourdes.gomez@uclm.es (L.G.-G.); 2Departamento de Química Inorgánica, Orgánica y Bioquímica, Facultad de Farmacia, Universidad de Castilla-La Mancha, C/José María Sánchez Ibáñez s/n, 02008 Albacete, Spain; Joaquinc.garcia@uclm.es; 3Regional Center for Biomedical Research (CRIB), Universidad de Castilla-La Mancha, C/Almansa 13, 02008 Albacete, Spain

**Keywords:** nanotechnology, cosmetic, chitosan, essential oils, antimicrobial, preservative agents, nanoparticles

## Abstract

The current status of controversy regarding the use of certain preservatives in cosmetic products makes it necessary to seek new ecological alternatives that are free of adverse effects on users. In our study, the natural terpene thymoquinone was encapsulated in chitosan nanoparticles. The nanoparticles were characterized by DLS and TEM, showing a particle size of 20 nm. The chemical structure, thermal properties, and release profile of thymoquinone were evaluated and showed a successful stabilization and sustained release of terpenes. The antimicrobial properties of the nanoparticles were evaluated against typical microbial contaminants found in cosmetic products, showing high antimicrobial properties. Furthermore, natural moisturizing cream inoculated with the aforementioned microorganisms was formulated with thymoquinone-chitosan nanoparticles to evaluate the preservative efficiency, indicating its promising use as a preservative in cosmetics.

## 1. Introduction

Microorganism contaminants are one of the most usual causes of alteration in cosmetic products (CPs) due to over-exposition to atmospheric oxygen [1]. When the CPs are opened for daily use, the microorganisms present in the atmosphere make contact with the cosmetic formulation, producing problems such as the appearance of mold, separation of phases of the emulsions, loss of viscosity, change in aroma, or rancidity of fats [2]. Moreover, the presence of pathogenic microorganisms in CPs has been associated with the development of some adverse effects, such as dermatitis, irritation, or peeling as well as conjunctivitis, asthma, urticaria, angioedema, or pneumonia [3].

The rising CP market, which is projected to reach $429.8 billion by 2022, is linked to an increased consumption of long shelf-life products. The cosmetic industry uses several compounds to prolong the period of use in cosmetics. One of the most common approaches is the incorporation of preservative agents, i.e., substances that inhibit the growth of microorganisms and counteract the generation of reactive oxygen species and oxidation processes [4]. Some cosmetic preservatives such as parabens, triclosan, benzalkonium chloride, formaldehyde, phenoxyethanol, and chlorphenesin produce adverse effects in humans such as DNA damage, antiandrogenic activity, cytotoxicity and genotoxic effects on human lymphocytes, cytotoxicity in meibomian glands, risk of cancer, and allergic reactions and also produce environmental and animal toxicity [5,6]. Due to these results, the traditional use of preservatives in cosmetics is leading us to search for new “eco-friendly” and efficient alternatives.

Essential oils (EO) are volatile compounds present in some aromatic plants and spices. Terpenes and isoprenoids are compounds derived from isoprenes and are the main component of several EOs that are widely used in medicine due to their varied biological and pharmacological activity, including high antimicrobial and antioxidant properties [7]. Thymoquinone (TQ) is the most abundant constituent of the volatile oil of *Nigella sativa* seeds, contributing to most of the properties of *N. sativa.* Different pharmacological properties of TQ have been previously described, e.g., immunomodulatory [8], anti-histaminic [9], anti-tumor [10], hepatoprotective [11], gastroprotective [12], nephroprotective [13], neuroprotective [14], antioxidant [15], and antimicrobial activity [16]. However, its high volatility and easy degradation can make its applications difficult.

The encapsulation of terpenes in different raw materials can improve the stability of compounds and increase their therapeutic efficacy. One of the most promising approaches is through the use of nanotechnology, with the encapsulation of terpenes in different formulations with a range of sizes between 1–1000 nm, thus improving their effect and stability [17].

Chitosan (CH), a linear polysaccharide consisting of 1, 4-linked glucosamine and *N*-acetylglucosamine, is a natural polymer derived through deacetylation of chitin in an alkaline media, with chitosan being the major component in shrimp and other crustaceans’ cuticular exoskeletons [18]. CH is one of the most popular polysaccharides used as a raw material to encapsulate active compounds such as essential oils, terpenes, and several drugs due to its structure-forming, biocompatibility, and high stability [19,20]. Due to its versatile and easy manipulation, several formulations of chitosan have been formulated such as gel beads [21], polyelectrolyte complex-based hydrogels [22], microparticles [23], and nanoparticles, among others. The encapsulation of TQ into chitosan nanoparticles has been tested in different medical areas [14], where the TQ encapsulation is evaluated in radioiodinated folic acid-chitosan nanoparticles in order to target ovarian cancer, showing lower cytotoxicity in healthy cells [24]. Encapsulation of TQ in chitosan nanoparticles coated with polysorbate 80 showed an improvement in the antidepressant effect of terpene in Wistar rodents [25]. Moreover, in the cosmetic industry, CH is widely applied in CPs as an antioxidant, emulsifying agent, and skin protective agent in chitosan form or in different chitosan forms such as carboxymethyl chitosan [26].

Thus, the present study shows, for the first time, the assessment of TQ encapsulated in chitosan nanoparticles (NPCH-TQ) as a cosmetic preservative agent through the evaluation of antimicrobial activity in natural formulated moisturizing cream.

## 2. Results

### 2.1. Particle Size and Z Potential of NPCH and NPCH-TQ

Photon correlation spectroscopy or dynamic light scattering (DLS) is a technique based on the scattering of a laser beam of a given wavelength by particles or macromolecules in a liquid medium due to their Brownian motion, which is widely used to determine particle size and surface charge. The results presented in Table 1 show an average range of nanoparticle size from 48.6 to 65.0 nm, showing an increase in size from non-encapsulated nanoparticles due to the incorporation of terpenes into raw material. The results of the polydispersity index (PDI) show values below 0.5, achieving PDI values of 0.2 in the formulations of ratios 1:0.75 and 1:1. All nano-formulations present positive Z values from +23.9 up to +49.8 mV due to the use of chitosan as a raw material. The results presented in Table 1 show a decreasing trend in Z values related to the amount of TQ encapsulated in nanoparticles.

### 2.2. Successful Encapsulation of Thymoquinone in Chitosan Nanoparticles

Encapsulation efficiency and efficient loading are parameters related to the capacity of encapsulation of drugs by different raw materials and the quality of formulation. These parameters mainly depend on the type of raw material, drug polarity, and approach used in nanoparticle preparation. The results presented in Table 1 show different EE% and EL% obtained by the different ratios used to obtain NPCH-TQ. The EE% values are between 88.2% and 94.5%, showing the maximum value in the 1:0.75 ratio. The maximum EL% value reached 50.7% in the 1:1 ratio corresponding to the formulation with the highest amount of initial thymoquinone used.

### 2.3. Morphology Characterization of Nanoparticles

Transmission Electron Microscope (TEM) is a common microscope technique to determine the surface properties and morphology characteristics of nanoparticles and nano-formulations [27]. Figure 1 shows two different micrographs which correspond to (A) NPCH and (B) NPCH-TQ 1:1. Both images show nanoparticles with non-defined surfaces, clearly trending toward forming aggregates or gel-like structures. All nanoparticles have a size smaller than 50 nm with approximately 20 nm with narrow size distribution.

### 2.4. Chemical Structure of NPCH-TQ

The chemical composition of NPCH-TQ was determined by identification of the principal functional groups through the FTIR technique. The attenuated total reflectance (ATR) accessory allowed us to identify the main functional groups present directly from the nanoparticles or essential oils without the need to use matrices. FTIR is typically employed to investigate the interaction between functional groups. However, this technique allows us to verify the existence of functional groups whose vibrational modes are characteristic. FTIR spectra of TQ, NPCH, and NPCH-TQ are shown in Appendix A. In the TQ spectrum, the signals corresponding to the stretching vibrations of the most important functional groups are observed. The most intense signal in the spectrum is broadband at 1636 cm^−1^, which corresponds to the overlap of two vibrations, the C=O double bond strain and the C=C double bond strain. The conjugate system between the C=O and C=C of the TQ makes these vibrations overlap in the spectrum. Moreover, intense bands corresponding to the sp^3^ carbon C-H bonds (2969 cm^−1^) and the C-O bond strain (1249 cm^−1^) are observed. The NPCH and NPCH-TQ spectra are very similar since their content is mainly chitosan. A broadband between 3750 cm^−1^ and 2500 cm^−1^, associated with the stretch of the O–H and N–H bonds, was clearly observed. As shown in the gravimetric analysis, chitosan nanoparticles were hygroscopic and had a great tendency to absorb water. This signal includes both chitosan O–H structure and water in its different strengths due to the hydrogen bonds formed. From the spectra, additional characteristic peaks of chitosan were observed at 2916 cm^−1^ and 2864 cm^−1^ due to C–H bond stretching of sp^3^ carbon, 1634 cm^−1^ due to carbonyl stretching C=O of the amide, 1533 cm^−1^ due to N–H bending of the chitosan, and 1066 cm^−1^ due to the tension of the different C-O bonds. Because the most important TQ signals overlap with important bands in the NPCH, it is difficult to appreciate significant changes, although variations in signal intensities indicate the presence of TQ in the nanoparticles [28].

### 2.5. Thermal Properties of NPCH-TQ

Thermo Gravimetric Analysis/Differential Scanning Calorimetry (TGA/DSC) was used to evaluate the absorbed material, thermal stability, and decomposition temperatures of the TQ, NPCH, and NPCH-TQ. TQ is a volatile yellowish solid. Appendix A shows the combination of TGA and DSC for TQ. In DSC, an endothermic peak is observed at 45 °C (119 J/g) which correlates with the loss of virtually all mass in TGA. This indicates that the compound is volatile and sublimes at this temperature. Previously, we showed that chitosan nanoparticles have two degradation steps, the initial step starting from the very beginning of the experiment and ending at 100 °C, and the second one starting at 170 °C up to 333 °C (Appendix A). The first transition corresponds to a 3% mass drop and is attributed to the loss of adsorbed/bound water/moisture vaporization [28]. The degradation of the pure chitosan biopolymer resulted in a 60% weight loss in the second stage. This has been found previously for various chitosan polymers, where the amount of moisture and the range of breakdown temperatures are dependent on the chitosan polymer’s molecular weight. A similar trend is observed in the NPCH-TQ thermogram (Appendix A), coinciding in the temperature ranges. In the second transition, the mass loss was 58%, matching with the degree of NPCH decomposition [28]. In first transition, the mass loss is considerably higher, reaching 9.4% and, even though the water absorbed by both particles may differ, this may indicate that this 9.4% loss could correspond not only to water but also to TQ loss. These data were confirmed by comparing the endothermic peaks between 25 °C and 100 °C. There is a considerable difference between the NPCH and NPCH-TQ evaporation enthalpies. For NPCH, an enthalpy of 145 J/g has been determined, while NPCH-TQ shows a higher value, reaching 274 J/g, which indicates that TQ-loaded nanoparticles need more energy to evaporate both water and TQ.

### 2.6. In Vitro Drug Release of NPCH-TQ

Drug release studies can offer information concerning nanoparticle behavior to release its encapsulated drugs. In this study, the pH of the buffer release medium was chosen according to the nanoparticle application. In this case, the pH of the CPs was close to 6. The results presented in Figure 2 show a triphasic release profile of TQ, achieving a maximum of 62% of TQ release at 28 days. Nanoparticles showed a slow initial release reaching 11% TQ in the first 24 h. Then, the release of TQ was governed by diffusion and degradation of the polymer matrix showing a slow-release profile due to the high hydrophobicity of TQ, resulting in the release mechanism being dependent on the nature of the raw material.

### 2.7. DPPH Scavenging Activity of Free TQ and Nano-Formulations

To determine the antioxidant properties of nano-formulations, we used the DPPH scavenging method. DPPH was chosen for its simplicity since it is one of the few organic radicals with nitrogen atoms in its structure. This structure confers stability because of the delocalization of an unpaired electron on the molecule. This delocalization also causes an intensification in the purplish color characteristic of the radical, which in an ethanolic medium absorbs at 515 nm. This purplish coloration of the solution is attenuated in the presence of an antioxidant that can donate or transfer a hydrogen atom, giving a yellowish color due to the reduced form of DPHP-H [5]. Figure 3 presents the % of DPPH scavenging of free TQ, NPCH and NPCH-TQ on the range of concentrations between 0.125 and 1 of tested material. NPCH did not show antioxidant properties against DPPH in the concentration range tested. Free TQ shows a maximum of 83.6% of inhibition of DPPH at 1 mg/mL of TQ, higher than NPCH-TQ, which showed a maximum inhibition of 29.3% at 1 mg/mL of total material corresponding to 0.48 mg/mL of pure TQ.

### 2.8. Antimicrobial Evaluation of Free Terpene and Its Nano-Formulation

The microorganisms used to check the activity in preservative agents in CPs are *P. aeruginosa*, *E. coli*, *S. aureus*, *A. brasiliensis*, and *C. albicans*. The broth microdilution method is a widely used technique to evaluate the antimicrobial properties of volatile compounds such as essential oil and terpenes [17,29]. The Minimum Inhibitory Concentrations (MICs) after 24 h of treatment for bacteria and 48 h for fungi are presented in Table 2. The minimum inhibitory concentrations (µg/mL) of different treatments are presented in Table 2. NPCH did not display an increase in the antimicrobial effect with respect to the control treatments against tested microorganisms, showing the same MICs in *E. coli* and *P. aeruginosa* (1000 µg/mL) and lower MICS against *S. aureus*, *C. albicans,* and *A. brasiliensis* (>1000 µg/mL). The encapsulation of TQ in chitosan nanoparticles resulted in improved antimicrobial activity reaching 292 µg/mL, 417 µg/mL, and 333 µg/mL in *E. coli*, *P. aeruginosa,* and *S. aureus*, respectively. However, the NPCH-TQ showed lower MICs than free TQ against *C. albicans* and *A. brasiliensis*, obtaining 250 µg/mL and 500 µg/mL, respectively.

### 2.9. Evaluation of NPCH-TQ as a New Preservative Agent in Cosmetic Products

In order to evaluate the effect of NPCH-TQ as new a preservative agent in cosmetic products, we formulated a natural moisturizing cream with the ingredients shown in Table 3. After formulation, control cream and NPCH-TQ cream were both inoculated with common contaminant microorganisms in CPs as recommended by UNI EN ISO 11930:2012, detailed in Section 4.8. At different intervals of days, 1 g of the creams was inoculated in different culture media, and the UFC/g was counted for each microorganism. The results of the antimicrobial activity of NPCH-TQ in the formulated cream are presented in Figure 4. The control cream did not display any preservative activity against all tested microorganisms, showing an increasing trend over the days in all bacteria and fungi reaching 10^8^ UFC/gr for *P. aeruginosa* and *E. coli* and 10^7^ for *S. aureus*. Preserved cream with NPCH-TQ displayed high antibacterial activity against *E. coli* and *S. aureus,* reducing the UFC/g to zero. Meanwhile, in *P. aeruginosa*, NPCH-TQ showed bacteriostatic properties during the time of the experiment, maintaining stable CFU/g. The antifungal effects of NPCH-TQ in formulated cream against *A. brasiliensis* and *C. albicans* are shown in Figure 4 revealed a total reduction in the number of UFC/g to zero.

## 3. Discussion

The main objective of this work is to develop a new alternative to conventional preservative agents based on green nanotechnology through the encapsulation of the natural terpene thymoquinone in chitosan nanoparticles. There is a lack of studies on TQ encapsulation in chitosan formulations; studies have focused mainly on the treatment of breast adenocarcinoma [30], ovarian cancer [24], nose-to-brain targeting [31], and depression treatments [25], among others.

In this study, we obtained four types of nano-formulations of NPCH-TQ through the utilization of different ratios of NPCH:TQ, where the particle size measured by the DLS technique did not show too many differences among obtained nano-formulations. The particle size ranged from 48.6 to 65.0 nm. However, the TEM micrograph presented in Figure 1 shows nanoparticles with a size close to 20 nm. Other works obtained the same trend with terpenes encapsulated in chitosan nanoparticles, obtaining different data regarding particle sizes in DLS and TEM. This was the case in the study performed by Woranuch et al., where chitosan-eugenol nanoparticles showed an average particle size of 683 nm by DLS and close to 100 nm by TEM due to the swelling and aggregation effect of chitosan in water solutions [32]. All NPCH-TQ revealed a positive surface charge with a decreasing trend related to the incorporation of TQ into the polymer matrix. The same pattern is shown in the encapsulation of other volatile compounds such as garlic essential oil in chitosan nanoparticles with a Z potential range from +45.8 to +19.8 mV [28]. The positive surface charge of nanoparticles is associated with increased antimicrobial activity due to their ease of interaction with the wall of microorganisms [17]. Our nano-formulations showed a successful encapsulation of TQ in all tested ratios, with a maximum of 94.5% of EE% and 50.7% of EL for the ratio 1:1. Other works with chitosan-TQ nanoparticles coated with polysorbate 80 reported an EE% of 85.6% and 16.26% of EL% [25]. The encapsulation of TQ has also been explored in other polymeric raw materials such as PLGA, showing EE% values of 79.9% [33].

The release profile in long-term experiments gives us an idea of how the nanoparticles release the encapsulated compound with a suitable pH for CPs as well as the release profile during 28 days of storage. NPCH-TQ showed a triphasic release profile with a maximum release of 62% at 28 days. The insolubility of TQ in water results in a slow burst release and diffusion phase showing a slower release profile than other water-soluble drugs such as doxorubicin encapsulated in chitosan nanoparticles, which reached 100% of release at 170 h [34]. Another work in which TQ was encapsulated in pegylated poly-lactide co-glycolide nanoparticles (TF-PEG-PLGA-TQ NPs) for the treatment of non-small cell lung carcinoma achieved a maximum of 50% cumulative release of TQ in 24 h at pH 7.4, showing a faster release than NPCH-TQ due to the more hydrophilic nature of PEG-PLGA, facilitating the release of the hydrophobic compound from raw materials [35]. More hydrophilic terpenes, such as eugenol, showed a faster release than the TQ profile encapsulated in chitosan nanoparticles, with a maximum release of 73% at 36 h [36].

The study of antioxidant properties of NPCH-TQ is suitable to increase the usefulness of this nano-formulation in CPs. NPCH-TQ showed a maximum DPPH scavenging of 29.3% at 1 mg/mL of total material, lower than other essential oil encapsulated in chitosan nanoparticles such as clove essential oil, which showed a maximum value of 71% at the same concentration. However, this low value might be due to the fact that at DPPH scavenging measure time (30 min) and based on the release profile shown in Figure 2, the TQ released to the medium only reached 3.3% of total encapsulated TQ, and the antioxidant properties were due mainly to the presence of TQ and not to the chitosan as shown in Figure 3. Another nano-formulation with eugenol encapsulated in chitosan nanoparticles showed a maximum of 70% inhibition of DPPH at 4.0 mg/mL. Meanwhile, carvacrol encapsulated in the same nano-formulation achieved close to 40% of DPPH inhibition [37].

The encapsulation of TQ in chitosan nanoparticles resulted in an improvement of antibacterial activity in the bacteria tested. The antibacterial activity of NPCH-TQ was higher against *E. coli* than *S. aureus* and *P. aeruginosa* with 292 µg/mL, 333 µg/mL, and 417 µg/mL, respectively. NPCH-TQ displayed a higher antibacterial activity against *E. coli* and *S. aureus* than other nanoparticles made of carvacrol and eugenol-chitosan nanoparticles, which showed MICs values of 0.5–1 µg/mL for the same bacteria [37]. However, the encapsulation of TQ in chitosan nanoparticles was not associated with an increase in the antifungal activity of TQ. TQ nanoparticles obtained through the ball milling method obtained lower MICs with values of 160 µg/mL against *C. albicans* [38]. The same trend was observed in *A. brasiliensis* after NPCH-TQ with an MIC value of 500 µg/mL. Garlic essential oil encapsulated in chitosan-TPP nanoparticles via crosslinking showed different MIC values against two different Aspergillus species, *A. niger* and *A. versicolor*, with 0.37 µg/mL and 3.33 µg/mL, respectively [28].

To confirm the preservative effect of NPCH-TQ in naturally formulated cream, we evaluate the antimicrobial activity in contaminated cream with the microorganisms described above. After 28 days of storage, non-preserved cream (control cream) shows an increase in total UFC/gr in all tested microorganisms, reaching an increase of 3 logs in *E. coli*, *P. aeruginosa*, and *A. brasiliensis*. However, the synthesized NPCH-TQ displayed a total reduction in UFC/gr in *E. coli*, *S. aureus*, *C. albicans*, and *A. brasiliensis* and a maintenance of the total UFC/gr of *P. aeruginosa* related to the lower activity of NPCH-TQ against this microorganism. These results confirm the preservative activity of NPCH-TQ in formulated creams. Other works with biogenic silver nanoparticles from *Iris tuberosa* aqueous extract confirm the role of nanotechnology to be used as a platform to obtain new preservative agents [5].

In conclusion, the results shown in this work present a new and effective preservative agent based on the encapsulation of a terpene in chitosan nanoparticles to satisfy the regulations and the growing demands of the consumers of natural, green, and sustainable products. This approach achieves a sustained and controlled release of TQ, allowing for a greater stabilization of the terpene and a higher antimicrobial activity over time and offering a long-lasting, effective, and preservative effect in natural cosmetics. However, additional experiments are necessary to support the use of our nanoparticles in humans before commercialization.

## 4. Materials and Methods

Low molecular weight chitosan (CH) (50–190 kDa) with a 75–85% degree of deacetylation, tripolyphosphate (TPP), 3-(4-dimethylthiazol-2-yl)-2 5-diphenyltetrazolium bromide (MTT), thymoquinone (TQ), and all the solvents were supplied by Sigma-Aldrich (Madrid, Spain). Microorganisms were purchased from the American Type Culture Collection (Manassas, VA, USA), namely *E. coli* (ATCC25922), *P. aureginosa* (ATCC27853), *S. aureus* (ATCC 6538), *C. albicans* (ATCC 10231), and *A. brasiliensis* (ATCC16404).

### 4.1. Preparation of Loaded and Unloaded Thymoquinone-Chitosan Nanoparticles (NPCH-TQ)

#### 4.1.1. Formulation of NPCH

Chitosan nanoparticles (NPCH) were formulated through the ionic-gelation method described by [28]. Briefly, CH solution at 0.2% was prepared by dissolving CH flakes in acetic acid at 1% under continuous stirring overnight. Then, 50 mL of CH solution was mixed at 1000 RPM in a 1% Tween 80 solution and heated to 50 °C. Finally, TPP aqueous solution at 0.2% was added dropwise at 2 mL/min under continuous stirring to induce the ionic gelation to form the nanoparticles. Afterward, agitation was carried out at 700 RPM for 40 min. The nanoparticles were collected after centrifugation at 15,000 RPM for 20 min at 4 °C and subsequently washed several times with mQ water. The nanoparticle suspension was frozen at −80 °C and freeze-dried for 48 h at −50 °C (LyoQuest-85/208 V 60 Hz, Teslar).

#### 4.1.2. Formulation of NPCH-TQ

Encapsulation of TQ into chitosan nanoparticles (NPCH-TQ) was formulated in a two-step process. Firstly, oil-in-water emulsification (o/w) was carried out, followed by the ionic gelation method as described above. Briefly, 50 mL of previously prepared CH solution was mixed at 1000 RPM in a 1% Tween 80 solution and heated at 50 °C. Subsequently, different amounts of terpene, previously preheated at 50 °C to form the liquid state to perform different ratios of CH:terpenes (1:0, 1:0.25, 1:0.5, 1:0.75, and 1:1 *w*/*w*), were added dropwise under continuous agitation and emulsified at 1500 RPM during 10 min at room temperature. Finally, TPP aqueous solution at 0.2% was added dropwise at 2 mL min^−1^ under continuous stirring to induce the ionic gelation. Afterward, agitation was carried out at 700 RPM for 40 min. The nanoparticles were collected after centrifugation at 15,000 RPM for 20 min at 4 °C and subsequently washed several times with mQ water to eliminate unencapsulated terpenes. The nanoparticle suspension was frozen at −80 °C and freeze-dried for 48 h at −50 °C (LyoQuest-85/208 V 60 Hz, Teslar).

### 4.2. Determination of Encapsulation Efficiency and Loading Efficiency of Thymoquinone-Chitosan Nanoparticles (NPCH-TQ)

#### Determination of Encapsulated TQ in Nanoparticles

To determine the total amount of TQ encapsulated into chitosan nanoparticles, 2 mL acetonitrile was added to 5 mg of TQ-NPCH, and suspension was sonicated for 1 h at room temperature and left overnight with the solvent. Then, the supernatant was filtered through 0.45 µm Millipore filter to measure in HPLC. The mobile phase employed was acetonitrile:water 70:30 and the amount of TQ was measured at 254 nm [39].

Loading capacity (LC) and encapsulation efficiency (EE) of terpenes were calculated according to the following equations (Equation (1), Equation (2)):LC% = (weight of encapsulated terpene (mg)) ÷ (weight of total (terpene encapsulated + scaffold weight) (mg)) × 100 (1)
EE% = (weight of encapsulated terpene (mg)) ÷ (weight of terpene feeding (mg)) × 100(2)

### 4.3. Instrumental Characterization of Nanoparticles

#### 4.3.1. Particle Size Analysis

Characterization of nano-formulations (size, zeta potential, and polydispersity index (PDI)) was determined by dynamic light scattering (DLS) using a Zetasizer (3000 HSM Malvern Ltd., Madrid, Spain) with the following specifications: chitosan refractive index (IR) of 1.700, absorption index 0.010, and water solvent RI: 1.33, with a viscosity of 0.8872 cP. Measurements were performed in triplicate.

#### 4.3.2. Chemical Analysis of Nanoparticles

R spectra were recorded on an attenuated total reflectance-Fourier transform infrared (ATR–FTIR) spectrophotometer (VARIAN 640-IR with a Pike Diamond/KRS-5 HS Performance Crystal Plate), and the main peaks were given in cm^−1^. ATR allows us to use the samples directly in a solid or liquid state without the need of KBr or Lugol’s iodine matrix. Specifically, for NPCH and NPCH-TQ, 20 mg of nanoparticles was powdered in a mortar, and the thin solid was placed on the diamond plate and pressed until a homogeneous pellet was obtained. For TQ, being liquid, a drop of approximately 200 μL was placed on the plate, and the tip was placed in such a way that the surface tension of the drop covered the diamond plate homogeneously. 256 scans were acquired at an instrument resolution of 1 cm^−1^ over the spectral range between 650 and 4000 cm^−1^ owing to the frequency cutoff of the ATR–FTIR internal reflection element (IRE) used.

#### 4.3.3. Thermal Properties of Nanoparticles

The thermal decomposition mechanisms were determined on a thermogravimetric analyzer (TGA Q20, TA Instruments) fitted with a standard platinum pan. The differential scanning calorimetry (DSC) experiments were carried out using a DSC Q50 system (TA Instruments) equipped with a standard aluminum pan with 10 °C/min increasing heat rate (30–320 °C) to investigate the thermal stability of pure TQ, NPCH, and NPCH-TQ. A sample of indium was used as reference. In all cases, samples of about 3 mg were heated at a 10 °C min^−1^ rate under nitrogen atmosphere.

#### 4.3.4. Morphology Studies of Nanoparticles

High-resolution electron microscope images of NPCH-TQ were obtained on a Jeol JEM 210 TEM microscope operating at 200 kV and equipped with an Oxford Link EDS detector. The resulting images were analyzed using Digital Micrograph™ software from Gatan.

### 4.4. In Vitro Release Studies of NPCH-TQ

One milligram of lyophilized nanoparticles was sealed in a dialysis membrane (molecular weight cut off 3500 Da) and suspended in 10 mL of phosphate-buffered saline at optimum pH in CPs (pH 6) in continuous stirring at 200 rpm to ensure homogeneity. At certain intervals of incubation time at 37 °C, 3 mL of release medium was removed to evaluate the amount of released terpene and replaced with 3 mL of fresh medium. The concentration of released TQ was determined with a spectrophotometer at wavelengths of 258 nm. Terpene release was performed in triplicate.

### 4.5. 2,2-Diphenyl-1-Picrylhydrazyl (DPPH) Radical Scavenging Activity

FRS, free radical scavenging activity, was determined as described previously [5]. Briefly, 0.5 mL for each concentration (1 mg/mL, 500 µg/mL, 250 µg/mL, and 125 µg/mL) of synthetized NPCH-CAR, NPCH-EUG, and free terpenes was mixed with 0.1 mM ethanolic DPPH radical solution (1.5 mL) and then mixed and kept in the dark at room temperature for 30 min. The absorbance of the solution was measured at 517 nm. The FRS was calculated by % = (A0 − A1/A0) × 100, where A0 is absorbance at Time = 0 and A1 is the absorbance after 30 min.

### 4.6. Antimicrobial Assay

Different ratios of CH:TQ (1:0.25, 1:0.5, 1:0.75, and 1:1) were evaluated in order to perform the optimal formulation. For antimicrobial analysis, the NPCH:TQ ratio 1:1 was chosen to perform the assay because this ratio has better loading efficiency values than the other ratios (50.7%) plus similar sizes and PDIs.

The antimicrobial activity and minimum inhibitory concentration (MIC) of terpenes and nanoparticles were tested against the most common pathogenic microorganisms in cosmetics and those that the UNI EN ISO 11930:2012 recommend for preservative efficacy evaluation. Antimicrobial activity of nanoparticles against *P. aeruginosa*, *E. coli*, *S. aureus*, *A. brasiliensis*, and *C. albicans* was tested using the broth microdilution method [17,40]. Stock cultures were prepared from Culti-Loops ™ (Sigma-Aldrich, Madrid, Spain) in Nutrient Broth (NB) and Potato Dextrose Broth (PDB) at 37 °C. Standardized inoculum was then created by dilution in Müller–Hinton medium to a final density of 0.5 McFarland units by densitometer McFarland type DEN-1B (Biosan, Riga, Latvia). TQ, NPCH-TQ, and NPCH were tested in concentrations of 1000 µg/mL to 15.6 µg/mL. Gentamicin (for bacteria) and Tebuconazole (for mold and yeast) were used as standards. After treatment, plates were incubated 24 h at 37 °C for bacteria and 48 h at 30 °C for yeast and fungi.

### 4.7. Moisturizing Cream Formulation

Two types of moisturizing creams were formulated: control cream without any preservative agent and NPCH-TQ with the addition of nanoparticles on formulated cream. The composition of both creams is shown in Table 3 Oil phase (B) and water phase (A) were preheated at 70 °C to achieve the fusion of oils and waxes present in this phase. Then, B was added slowly under agitation in a homogenizer at 3000 RPM to form an oil/water emulsion. Upon cooling, the thermolabile compounds (C) and preservatives (NPCH-TQ) at a dose of 500 µg/mL (maximum of MIC value) were added to the cooled cream under continuous agitation.

### 4.8. Preservative Activity of AgNPs in Formulated Cream

The preservative efficacy of NPCH-TQ in the moisturizing creams was carried out comparing the antimicrobial activity of NPCH-TQ cream against non-preserved cream. Briefly, twenty grams of each cream (Control and NPCH-TQ) was diluted with a sterile NaCl solution at 0.9%. Then, each cream was inoculated with 10^5^ UFC/mL for bacteria, 10^4^ CFU/mL for yeast, and 10^3^ CFU/mL for mold. Contaminated creams were stored at room temperature for 30 days. After 2, 7, 14, and 28 days, one gram of each contaminated cream was diluted and spread in Petri dishes to count CFUs of bacteria. Yeast and mold contaminations were evaluated at 7, 14, and 28 days using the same methodology.

## Figures and Tables

**Figure 1 ijms-23-00898-f001:**
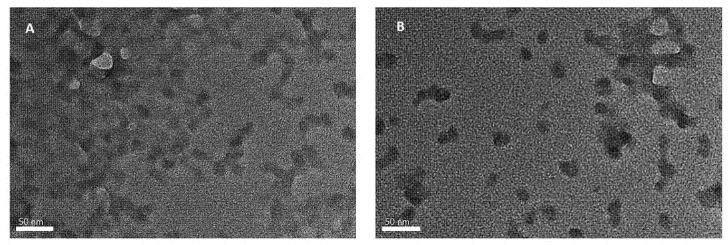
TEM micrograph of (**A**) NPCH and (**B**) NPCH-TQ 1:1.

**Figure 2 ijms-23-00898-f002:**
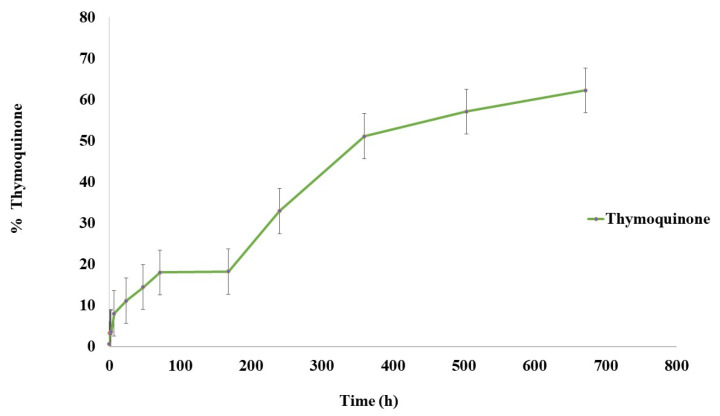
In vitro drug release of NPCH-TQ in PBS at pH 6.

**Figure 3 ijms-23-00898-f003:**
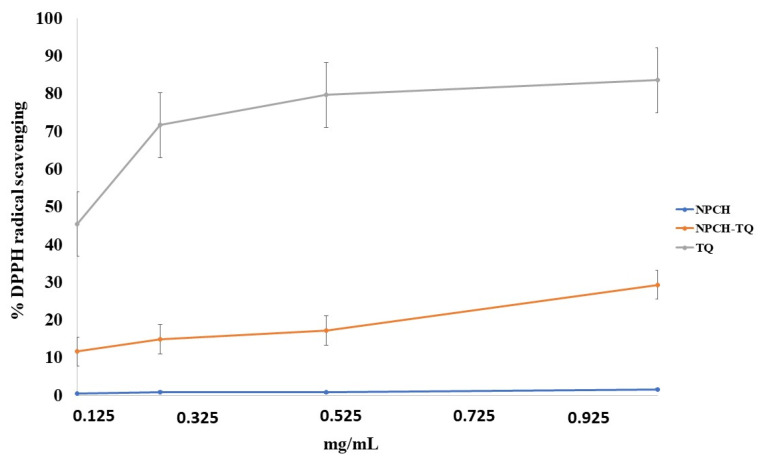
DPPH scavenging of free TQ, NPCH, and NPCH-TQ.

**Figure 4 ijms-23-00898-f004:**
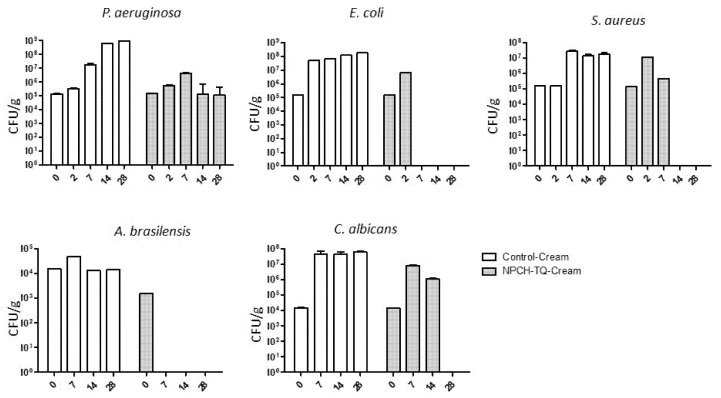
Preservative activity of control cream vs. NPCH-TQ.

**Table 1 ijms-23-00898-t001:** DLS measures, encapsulation efficiency (EE%), and efficient loading (EL%) of nano-formulations.

Formulation	Average Size (nm)	PDI	Z-Value (mV)	EE%	EL%
NPCH	48.6 ± 3.40	0.4 ± 0.02	+49.8 ± 0.75	-	-
NPCH-TQ 1:0.25	65.0 ± 1.40	0.4 ± 0.01	+35.8 ± 3.23	88.2 ± 6.39	44.8 ± 0.70
NPCH-TQ 1:0.5	57.5 ± 0.36	0.3 ± 0.01	+27.8 ± 1.13	93.2 ± 2.05	48.4 ± 1.14
NPCH-TQ 1:0.75	57.5 ± 0.33	0.2 ± 0.01	+25.3 ± 0.76	94.5 ± 1.94	48.8 ± 1.24
NPCH-TQ 1:1	63.4 ± 0.65	0.2 ± 0.01	+23.9 ± 0.58	90.6 ± 10.4	50.7 ± 8.70

**Table 2 ijms-23-00898-t002:** Minimum inhibitory concentrations (µg/mL) of control drugs (Gentamicin (*) and Tebuconazole (**)), TQ, NPCH, and NPCH-TQ.

Microorganism	Control MIC (µg/mL)	TQ (µg/mL)	NPCH (µg/mL)	NPCH-TQ (µg/mL)
*E. coli*	* 1000	1000	1000	292
*P. aeruginosa*	* 1000	>1000	1000	417
*S. aureus*	* 1000	>1000	>1000	333
*C. albicans*	** 250	333	>1000	250
*A. brasiliensis*	** 250	250	>1000	500

**Table 3 ijms-23-00898-t003:** Cream composition. A (Aqueous phase), B (Oil phase), and C (thermolabile compounds).

Ingredient	Control Cream (%)	NPCH-TQ Cream (%)
Water (A)	47	46.5
Vegetable glycerin (A)	10	10
Urea (A)	3	3
Glyceryl monostearate (B)	8	8
Argania spinosa kernel oil (B)	28	28
Allantoin (C)	0.4	0.4
Avena sativa extract (C)	3	3
Vitamin E (C)	0.5	0.5
Parfum (C)	0.1	0.1
NPCH-TQ (C)	-	0.5

## Data Availability

Not applicable.

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
