# Peer review of "Thymoquinone-Loaded Chitosan Nanoparticles as Natural Preservative Agent in Cosmetic Products"

_ijms, 2022, doi:10.3390/ijms23020898_

Round 1

Reviewer 1 Report

Dear authors, your article, Thymoquinone loaded chitosan nanoparticles as new “ecofriendly” preservative agent in cosmetic products, has an interesting topic and the research is thoroughly realised. The experiments used seem to be performed correct and the results were interpreted correctly.

The authors should explain in the manuscript the choice of microorganisms that were tested. How relevant are those in the context of the cosmetic use.

I recommend you to read it again with attention because you have some small error, most of them of editing, especially in Materials and Methods section. See for example on row 43, it should be “chlorphenesin”, and not “Chlophenesin”. On row 128, it should be a space between the value and the cm-1. In figure 2, use the decimal point. The same for figure 3.

Row 197, define what MIC is

Row 398, explain what is A0 and A1.

Row 352, use the journal’s style for formulas and equations.

Row 225, use g, and not gr. The authors should also correct the figures.

Author Response

Dear authors, your article, Thymoquinone loaded chitosan nanoparticles as new “ecofriendly” preservative agent in cosmetic products, has an interesting topic and the research is thoroughly realised. The experiments used seem to be performed correct and the results were interpreted correctly.

The authors should explain in the manuscript the choice of microorganisms that were tested. How relevant are those in the context of the cosmetic use.

Thank you very much for giving us the opportunity to resubmit our manuscript in consideration for publication in your journal. We also wish to thank all reviewers for their revisions and good comments which led to the improvement of the manuscript. We are submitting the revised version of our manuscript (ijms-1547035). We are describing the changes we have made to the manuscript in response the reviewer’s comments. All suggestions of reviewers were incorporated. We also asked a native English speaker to check the English level. In addition, we have responded to all questions asked by revierwer 1 

  • I recommend you to read it again with attention because you have some small error, most of them of editing, especially in Materials and Methods section. See for example on row 43, it should be “chlorphenesin”, and not “Chlophenesin”.

The required information has been included “Phenoxyethanol and Chlorphenesin produce adverse effects in humans such as DNA damage, antiandrogenic activity, cytotoxicity and genotoxic effects on human lymphocytes, cytotoxicity in meibomian glands, risk of cancer, allergic reactions and environmental and animal toxicity [5], [6]”

  • On row 128, it should be a space between the value and the cm-1. In figure 2, use the decimal point. The same for figure 3.

Done as suggested

  • Row 197, define what MIC is

Page 6, Row 198: The Minimum Inhibitory Concentrations ( MICs ) after 24 hours of treatment for bacteria and 48 hours for fungi are presented in Table 2.

  • Row 398, explain what is A0 and A1.

 The required information has been included “The absorbance of the solution was measured at 517 nm. The FRS was calculated by % = (A0 − A1/A0) × 100, where A0 is absorbance at Time = 0 and A1 is the absorbance after 30 min”.

  • Row 352, use the journal’s style for formulas and equations.

Done as suggested

  • Row 225, use g, and not gr. The authors should also correct the figures

Done as suggested

Reviewer 2 Report

The study by Niza et al titled Thymoquinone loaded chitosan nanoparticles as new “eco- friendly” preservative agent in cosmetic products” reports the fabrication of chitosan nanoparticles loaded with thymoquinone. The practical application of these nanoparticles is in using them as cosmetics antimicrobials. The authors performed some physico-chemical characterization and demonstrate the antimicrobial properties of the resulting particles using several bacteria. The authors make the following conclusion: “the results shown in this work present a never tested before alternative to conventional preservative agents based on the use of a terpene encapsulated in chitosan nanoparticles to be used as an” eco-friendly” preservative agent in cosmetic products. This approach achieves a sustained and controlled release of TQ allowing for a greater stabilization of the terpene and a higher antimicrobial activity over time, offering a long-lasting, effective, and safe preservative effect in natural cosmetics.”

I agree that this material can pose as an alternative. Terpenes indeed are supposed to demonstrate strong antimicrobial effects. As a result, I recommend this study to be published after some revisions

  • The authors claim that this preservative is “safe”. However, I see no results confirming the safety of the nanoparticles obtained. Therefore, the authors have to indicate that no safety studies were actually performed and downplay this claim.
  • Characterisation of the particles is quite mediocre. The authors used DLS, zeta-potential measurements and TEM. They state that “Transmission Electron Microscope (TEM) is one the most used microscope techniques to determine the surface properties and morphology characteristics of nanoparticles and nano-formulations”. This statement is rather incorrect. TEM is not the primary surface characterization technique. As evidenced in Figure 1 – no surface information is actually available. The images shown are of quite poor quality, scale bar is impossible to see. The spherical morphology claimed is not actually seen. I suggest the authors either provide better quality SEM and or AFM images, or re-write this section and critically assess their results.
  • I do not see any results on the colloid stability of these nanoparticles. How stable are they in water and in the cream?
  • What is the reason to call these particles eco-friendly? The authors did not actually investigate any effects on environment. It appears that the authors assume that this property comes from the raw materials used. However, it is not that easy and straightforward. It is quite likely that the paerticles might be toxic in some way. This needs to be corrected in the title and better explained in the text.
  • The literature report needs some update. Some recent relevant papers featuring the use of chitosan in similar polymer platforms for drug encapsulation and release are suggested:

https://doi.org/10.3390/coatings9020070

https://doi.org/10.3390/ma14010086

https://doi.org/10.3390/pharmaceutics12050455

After the revisions, the paper can be considered for publication.

Author Response

The study by Niza et al titled Thymoquinone loaded chitosan nanoparticles as new “eco- friendly” preservative agent in cosmetic products” reports the fabrication of chitosan nanoparticles loaded with thymoquinone. The practical application of these nanoparticles is in using them as cosmetics antimicrobials. The authors performed some physico-chemical characterization and demonstrate the antimicrobial properties of the resulting particles using several bacteria. The authors make the following conclusion: “the results shown in this work present a never tested before alternative to conventional preservative agents based on the use of a terpene encapsulated in chitosan nanoparticles to be used as an” eco-friendly” preservative agent in cosmetic products. This approach achieves a sustained and controlled release of TQ allowing for a greater stabilization of the terpene and a higher antimicrobial activity over time, offering a long-lasting, effective, and safe preservative effect in natural cosmetics.”

I agree that this material can pose as an alternative. Terpenes indeed are supposed to demonstrate strong antimicrobial effects. As a result, I recommend this study to be published after some revisions

We would like to thank the reviewer again for his/her full revision of our manuscript. We have changed the manuscript based on what reviewer suggested.

  • The authors claim that this preservative is “safe”. However, I see no results confirming the safety of the nanoparticles obtained. Therefore, the authors have to indicate that no safety studies were actually performed and downplay this claim.
  • Chitosan is widely used as an ingredient in cosmetic products producing an antioxidant effect, or as an emulsifying agent and skin protector among other uses. The regulatory agencies such as European agency through the Regulation (EC) No 1907/2006 (REACH) and Food and Drug Association (FDA) consider Chitosan an ingredient biocompatible and safety for human in cosmetic products (Polymers 2018, 10, 213; doi:10.3390/polym10020213) (https://ntp.niehs.nih.gov/ntp/htdocs/chem_background/exsumpdf/chitosan_508.pdf). On the other hand Thymoquinone and the essential oil obtained from, Nigella sativa, is considered as an antimicrobial agent used for wound healing, hyperpigmentation treatments, anti-inflammatory among others showing a safety profile for these uses  (https://doi.org/10.1016/j.jdds.2015.04.002) (https://doi.org/10.3390/nu10101369). We have followed the reviewer’s suggestion and we included in final conclusions the necessity of additional experiments to support the use of our nanoparticles in humans .
  • Characterisation of the particles is quite mediocre. The authors used DLS, zeta-potential measurements and TEM. They state that “Transmission Electron Microscope (TEM) is one the most used microscope techniques to determine the surface properties and morphology characteristics of nanoparticles and nano-formulations”. This statement is rather incorrect. TEM is not the primary surface characterization technique. As evidenced in Figure 1 – no surface information is actually available. The images shown are of quite poor quality, scale bar is impossible to see. The spherical morphology claimed is not actually seen. I suggest the authors either provide better quality SEM and or AFM images, or re-write this section and critically assess their results.

The reviewer is correct, and we appreciate the chance to make ourselves clearer. A new paragraph has been added “Transmission Electron Microscope (TEM) is  a common microscope techniques to determine  the surface properties and morphology characteristics of nanoparticles and nano-formulations [24]. Figure 1 shows two different micrographs which correspond to A) NPCH and B) NPCH-TQ 1:1. Both images show nanoparticles with non-defined surfaces, clearly trending to form aggregates or gel-like structures. All nanoparticles have a size smaller than 50 nm with approximately 20 nm with narrow size distribution”.

The image has been changed as suggested

  • I do not see any results on the colloid stability of these nanoparticles. How stable are they in water and in the cream?

We have checked the size and polydispersity index (PDI) of nanoparticles during the release studies to evaluate the stability of nanoparticles in aqueous suspension at pH 6 (suitable pH in cosmetic products). The measurements were performed at T1, T10, T15 and T 28 as showed in Table 1. After 28 days the NPs did not show significant differences in their mean size and PDI indicating good stability in aqueous media. Due to the galenic form of our cream with high density and multicomponent ingredients, we cannot isolate a suitable sample to test the stability of the nanoparticles.

Table 1: Size and polydispersity index (PDI) of nanoparticles, measured from day 1 to day 28.

Day

Size

PDI

1

63,4±0,65

0.2 ± 0.01

10

83,4±0,38

0.2 ± 0.01

15

79,6±0,52

0.2 ± 0.01

28

65,4±0,43

0.2   ± 0.01

  • What is the reason to call these particles eco-friendly? The authors did not actually investigate any effects on environment. It appears that the authors assume that this property comes from the raw materials used. However, it is not that easy and straightforward. It is quite likely that the paerticles might be toxic in some way. This needs to be corrected in the title and better explained in the text.

Both chitosan and thymoquinone are considered greener ingredients in cosmetic products in comparison to the traditional preservative systems (https://doi.org/10.3390/md17060369), we believe that our nanoparticles could be considered also a "green" and eco-friendly alternative at the tested doses due to its biodegradability and low impact in the environment (https://doi.org/10.1016/j.postharvbio.2018.01.014) (https://doi.org/10.1080/10942912.2017.1345935) (DOI: 10.3923/ajppaj.2017.53.70). In accordance with the reviewer’s comment, we have revised the manuscript title as follow: Thymoquinone loaded chitosan nanoparticles as natural preservative agent in cosmetic products

New paragraphs have been added

Page 7, line 230: The main objective of this work is to develop a new alternative to conventional preservative agents based on green nanotechnology through the encapsulation of the natural terpene thymoquinone in chitosan nanoparticles

Page 9, line 307. In conclusion, the results shown in this work present a new and effective preservative agent based on the encapsulation of a terpene in chitosan nanoparticles to satisfy the regulatory, and the growing demands by the consumers of natural, green and sustainable products.

  • The literature report needs some update. Some recent relevant papers featuring the use of chitosan in similar polymer platforms for drug encapsulation and release are suggested:

https://doi.org/10.3390/coatings9020070

https://doi.org/10.3390/ma14010086

https://doi.org/10.3390/pharmaceutics12050455

The requested references were added to the manuscript.
